# Use of Plantar Pressure Sensors to Take Weight-Bearing Foot Casts

**DOI:** 10.3390/s21227476

**Published:** 2021-11-10

**Authors:** Enrique Panera-Rico, José Manuel Castillo-López, Inmaculada Concepción Palomo-Toucedo, Fernando Chacón-Giráldez, Javier Ramos-Ortega, Gabriel Domínguez-Maldonado

**Affiliations:** Departamento de Podología, Universidad de Sevilla, 41010 Seville, Spain; epanera@us.es (E.P.-R.); jmcastillo@us.es (J.M.C.-L.); ipalomo@us.es (I.C.P.-T.); ferchacongi@us.es (F.C.-G.); jrortega@us.es (J.R.-O.)

**Keywords:** foot cast, foot mold, insole, foot orthosis, heel symmetry

## Abstract

Techniques of taking casts mainly rely not on the objectivity of the procedure, but on the experience and skill of the technician. The aim of this study was to demonstrate the efficiency of a technique of taking standing foot casts controlled via pressure sensors. In this way, we mean to objectivize the degree of correction. The study was carried out through 150 procedures on 50 feet of 29 patients. The value of the “Heel Symmetry Index” was calculated on three casts in three different situations of the same foot: A first cast in which the subject did not control the position of his/her foot; a second cast where manipulations corrected the foot’s pronator position; and a third cast with pressure sensors placed in the subject’s heel. This enabled the control and quantification of the pressure during the manipulation when taking the cast. The comparison of the “Heel Symmetry Index” in the different groups showed significant p-values of 0.05. Conclusion: The technique of taking casts controlled by pressure sensors achieved more equilibrated casts with a better symmetry index of the heel’s outline.

## 1. Introduction

In the last decades, the physical activity of the general population has progressively increased, therefore, more time dedicated to walking and running becomes absolutely necessary. Understanding biomechanics and foot function is essential to keep lower limb heath [1,2]. Foot orthosis has been traditionally used in clinical practice to manage some podiatric disorders in children and adults. Moreover, either to relieve pain or to improve biomechanical function of the lower limb, it is a common therapeutic option, particularly when a hyperpronation is diagnosed [3,4]. Some studies suggest that customized insoles can be useful in people with rheumatoid arthritis [5], Ehlers-Danlos or hypermobility syndromes and for the prevention of diabetic foot wounds [6]. In addition, it is an important part of the conservative treatment in sports lower extremity injuries and its prevention [7]. For a successful result, after clinical assessment of the patient, foot impressions the starting of custom-made foot orthoses manufacture, being considered a fundamental step because an improper process in the casting could cause a deficient foot orthoses [8,9,10,11].

There is a wide variety of foot casting techniques described, such as non-weight-bearing casting, semi-weight-bearing casting and weight-bearing casting. The traditional gold standard procedure is the negative impression of the foot morphology obtained in the prone or supine position, using plaster bandage also named plaster of Paris [1,9,12].

Foot weight bearing cast can be obtained on a depressible material, such as phenolic foam, in the standing position. The same material is used for taking semi-load cast when the patient is sitting on a chair [12,13]. In both cases, a three-dimensional footprint replication is obtained which is called negative mold. The positive mold can be acquired by filling it with liquid plaster [12,13]. Recent procedures, such as 3D scanning technology and laser scan, have not improved the accuracy and reliability of taking casts as compared to the conventional ones described [11,12,14].

A common method among professionals intends to maintain the subtalar joint (SJ) in a neutral position [8,15]. This does not mean that it has to be the most appropriate way of taking the cast. This maneuver is done by probing the head of the talus bone, a subjective technique by the practitioner, the same as achieving this position in a locked point in a neutral position [1]. This has made the maneuver controversial among different authors [2,9,15]. The technique is more complex when it is carried out with the patient standing, supporting the weight of his/her body, where it is more difficult to control the SJ’s neutral position.

In this case, the physician needs to apply an external manipulation of the patient foot to achieve the ideal position for the expected posture correction. We have not found literature recording of any reliable and measurable method, which can be reproduced with objective criteria, so that casting depends primarily on the skill and experience of the doctor for each method [11,15].

On the other hand, several studies are based on the evaluation of gait and foot position using wearable plantar sensors; however, they focus on kinetic or kinematic parameters of the human gait in normal or pathological conditions [16]. Other authors incorporate this instrument for monitoring the effect of the insoles while standing or walking [17]. This study proposes a novel and reproducible method to take foot weight-bearing casts, using a system of plantar sensors to guide the control of the practitioner’s manipulations, which allows positioning the foot according to the balance of plantar pressures, based on lateral and medial heel pressure.

## 2. Materials and Methods

### 2.1. Design and Sampling

The study is a pre-experiment, pre- and post-test design with a single intervention group with a convenience sampling [18] made up of 29 volunteers belongs to the Clinical Area of Podiatry of the University of Seville.

The participants included in the study met the following inclusion criteria: Over 20 years old (well consolidated bone development) and less than 50 years old to avoid aging related deformations. They were healthy subjects with values in the Foot Posture Index (FPI) greater or equal to 6 (pronated foot) in, at last, one extremity. Exclusion criteria were: Previous trauma or injuries that might affect the foot’s normal structure, musculoskeletal diseases, gait disorders, pregnancy, osteoarticular surgery of the foot, hallux valgus or digital deformations that could not be reducible [19]. 

Participants voluntarily signed an informed consent document after being informed about the characteristics of the study. 

The internal reliability was verified using 8 random feet from the total of the sample. To identify the intra-observational variability, three casts of those feet were taken in every three positions.

### 2.2. Recruitment

The members of the research team invited volunteers from the Clinical Area to participate in the study and they were informed about it. When the subject accepted, a brief anamnesis and basic physical examination was carried out to identify exclusion criteria. In case they did not exist, the FPI assessment was performed for each foot with the volunteer in standing relaxed position, on a podoscope. 

The FPI tool quantify the standing foot posture in three categories: supinated, normal o pronated, and it is based on the score (total range +12 to −12) obtained from the evaluation of 6 items with an individual range from +2 to −2: Talar head palpation, observation of the supra and infra lateral malleoli curvatures, calcaneal frontal plane position, observation of the medial prominence of tarsal-navicular joint, congruence of medial longitudinal arch and finally, abduction/adduction of the forefoot on the rearfoot. 

Only the feet with a FPI score greater than 5 points (pronated posture) were included in the study. Then, the participants signed the informed consent document and subsequently they went to the plaster room for the interventions.

### 2.3. Material

-Pressure measurement computer system WalkinSense^®^ with sensors.-Vacuum molds for taking the print.-A vernier caliper gauge or caliper normalized according to the standard DIN 862.-Articulated support arm for 2 kg Code RS387-0026.

### 2.4. Variables 

The “Intervention group”, with three possible values: Group without Correction, Group with Correction without Sensors and Group with Correction with Sensors were the independent variables of the study. The dependent variable was “Heel Symmetry Index”, which quantitatively values the symmetry of the heel’s outline in the frontal plane, indicating that the heel is centered and a perpendicular position related to the floor. 

Gender and age data were also registered.

### 2.5. Procedure

All the participants followed the same order and the three positions at the same time. No further repetitions were performed except for the feet included in the intra-rater reliability study. 

Taking the Cast in the Group without Correction

First, directly on the vacuum molds, a negative standing cast was taken in a calcaneal-relaxed posture (with no correction). The cast had an internal arc lowered and flattened by the midfoot pronation. Likewise, the heel zone in its medial part was more pushed down by the deformation of the valgus heel, giving an asymmetrical outline of the heel. 

Next, we filled the impression with plaster and once it had set, we emptied it, being the positive cast used for the study’s measurements.

Taking the Cast in the Group with Correction without Sensors

A second cast was taken with a manual correction maneuver, of the same foot with the subject in the same standing position. An external rotation of the distal third of the leg was performed by the practitioner’s hand. The opposite hand probed the head of the talus until perceiving that it was in a neutral position to correct the midfoot pronation.

The same as in the previous procedure, the cast was emptied and the positive cast was obtained. 

Taking the Cast in the Group with Correction with Sensors

The third cast was taken using the pressure sensors, stuck previously in three fixing points in the plantar heel: one in the middle line of the heel in its plantar surface and the other two at a centimeter, internal lateral and external lateral, from the central sensor (Figure 1).

The sensors were connected to the transmitter and this was in turn connected to the computer via Bluetooth^®^ to read the pressures in real time (Figure 2 and Figure 3).

In this way, the cast was taken carrying out the correction maneuver and visualizing the pressures for each sensor and in the same screen a graph represented the sensors in distinct colors, describing a line in the graph in real time and quantifying the pressure of each sensor. So, when the sensors quantified the same pressure, only a single coinciding line was perceived instead of three different lines.

The positive cast was then obtained as in the previous procedures. 

### 2.6. Method for Heel Symmetry Index Measurement

Having obtained three casts from the same foot, we cut each heel perpendicularly on the longitudinal axis of the foot (Figure 4) with a table-cutting saw, by the zone of maximum longitudinal and transversal curvature of the heel (Figure 5). This left us with its outline on the frontal plane.

The piece of the heel cut was colored to better visualize the curvature, using a distinct colour for each procedure to enhance distinguishing which group each one corresponds to.

Next, the frontal side of the piece was scanned on a scale (1:1). The outline of the curvature of the cut of the cast was visualized. We translated this to the AutoCAD^®^ 2009 computer program (Autodesk Inc, San Rafael, CA, USA), where the figure was scaled and the measurements shown in Figure 4 were done.

All these data were collected to do the operations through a mathematical formula of “Indexes based on the pairs of distances”. A quantity closest to zero or equal to zero was obtained. This was indicative of the symmetry of the curve, as the closer it is to zero, the more symmetrical the curve is:$$I^{\prime }=\overset{m}{\underset{k=1}{\sum }}{\left(h(x_{k}\right)-h(x_{-k}))}^{2}$$

The statistical analysis was carried out using the IBM SPSS Statistics 22 statistics packet for Windows^®^ (SPSS Science, Chicago, IL, USA). For the descriptive analysis, we calculated the number of subjects (*n*) as well as the corresponding percentage (%) for the sex variable. In the rest of the variables (age, cast without correction, cast with correction without sensors and cast with correction with sensors), we obtained the average values, the standard deviation (S.T.), the minimum, the maximum and the percentiles 25, 50 and 75 (P25, P50 and P75).

A level of confidence of 95% was taken into account for the inferential analysis, so the *p*-value was compared with a significance level of 5%.

We applied Friedman’s bidimensional analysis of variance by ranges for related samples after the Shapiro–Wilk test was applied. We carried out an analysis through Pearson’s correlation coefficient to study the relations between the quantitative variables.

For the study’s internal reliability, 8 feet were needed and 3 equal casts for each foot were taken in the 3 different described situations. We first determined the sample’s normality or non-normality using the Shapiro–Wilk test. Via the Friedman test, we checked the relation between the different measurements.

## 3. Results

A total of 150 procedures were performed from 50 feet of 29 volunteers, as eight unilateral feet were exclude because FPI score was less than six points. Of these, 16 belong to men (32%) and 34 to women (68%). The mean age was 24.7 years old, ±4.88 years, range 20 to 42 years old. The minimum sample size calculated was 40 feet, but to avoid experimental mortality, due to falls or deformations when obtaining the positive plaster, we included 50 ones. To check the study’s internal reliability eight feet were used so 72 procedures were performed, in addition (the minimum sample size calculated for this purpose was seven feet). 

### 3.1. Heel Symmetry Index

The “heel symmetry index” of the casts taken without correction, with correction without sensors and the casts with correction with sensors were the quantitative variables (Table 1).

We noticed that the means of Symmetry Index and the standard deviations followed the same decreasing tendency, in the three groups when more control position of the foot is applied. The mean data in the situation of taking the cast with correction with sensors were closer to the zero value. 

Percentiles 25, 50 and 75 showed the same decreasing tendency as well. The highest value percentile was 75 in the situation of taking casts without control without sensors, and the values being less than 0.068 and the closest to zero in the position with manual control with sensors information. 

Shapiro–Wilk normality test, indicated that the study variables did not show a normal distribution (Table 2).

Then, the Friedman test was applied to compare measurements of the variable “Symmetry Index”. The values of the variable studied are different, as the significance or *p*-values of the three casts compared by pairs were less than 0.05 (Table 3).

Figure 6 shows the values of the “Symmetry index” in the three groups of study. We emphasized that the highest values are found in the casts without correction, marked by a great dispersion of the data. The casts with correction with sensors, were the most homogeneous and less dispersed values, showing less variability.

### 3.2. Study of the Reliability

For the study of reliability, the same measure was taken three times for each foot in the three distinct suppositions for the eight subjects. Variation between the measurement of the casts without sensors in the three measurements is only 0.018. 

Shapiro–Wilk test for small samples verified that the distribution did not meet normality criteria, therefore non-parametric test was chosen. Friedman’s bidirectional analysis of variance by ranges for related samples was used to compare casts without sensors in the three measurements obtained (*p* = 0.093) and to compare casts with sensors in the three measurements (*p* = 0.882) 

In both cases, the three measurements can be considered similar, but in the case of the measurements without sensors this could not be stated with a confidence level of 90%, as the significance is 0.093. It can, therefore, be concluded that the measurements with sensors are more reliable at any confidence level admitted than the measurements without sensors.

## 4. Discussion

This study evaluated a reproducible method to take foot weight-bearing casts, using a system of plantar sensors to guide the control of the practitioner’s manipulations, based on lateral and medial heel pressure.

Volunteers included presented pronated feet, according to the Foot Posture Index (FPI) assessment [19,20,21], unlike other studies consulted that were done with people who did not suffer from any pathology [22,23,24,25,26]. We think that the results can be affected when no control is necessary, since casting techniques with and without weight bearing are evaluated. When the foot does not present any deformation to correct, the differences between the casts might be caused by the professional him/herself, who takes the cast precisely due to the procedure lacking objective references.

The foot’s position when taking the cast is a controversial topic, as it is also influenced by clinical experience. In this sense, some aspects of Root et al.’s [27] biomechanical model need a reconsidering of the biomechanical bases due to studies that question Root’s statement with respect to the STJ’s neutral position [23,24,25,26,27,28]. Other problems refer to the clinical identification of the subtalus’ neutral position as this method probes the talus head, which is a subjective procedure. Sobel [29] questions the subtalus’ neutral position when taking the cast. Referenced studies contradict Root’s theory concerning the neutral position and argue that taking the casts continues being more an art than a scientific discipline, perhaps owing to the lack of scientific control and rigor.

In this study, we decided to opt for taking the cast in the neutral STJ position. Analyzing the casts visually, the main difference that we noted was in the heel’s outline, as in a flat foot the convexity of the outline is displaced towards the external edge and is thus more pronounced and with the corrected foot it remained more symmetrical. This is why it was decided to analyze that outline and quantify its symmetry. This led us to value the novel parameter “Heel Symmetry Index” in its frontal plane, because this is a view that the professional takes much into account to value the cast’s position before making the plantar support.

The “Symmetry Index” is a parameter that is not mentioned in any study. According to the statistical results, the use of pressure sensors contributes objective references to obtain an equilibrated or symmetrical heel, as the division of equal pressures on the heel’s internal and external edges is controlled through the manipulation maneuver.

The study of the different casting techniques has been the aim of diverse works that try to reveal which technique is the most efficient. This is why they analyze various points and can thus objectify the differences between the casts [21,25,29,30]. In this study the “Symmetry Index” enables the differentiating of a cast of a flat foot from one in supination.

The statistics results evidenced that the use of pressure sensors helps to obtain more equilibrated casts. These are more morphologically similar to the casts made with plaster in non-weight-bearing position. As they have this visual reference, authors such as McPoil et al. [8], Someres et al. [25] and Laughton et al. [24] defend obtaining better results with this method than with phenolic foam. 

The statistical tests of reliability contributed satisfactory results with a confidence level of 0.882 for the casts with sensors and 0.093 for the casts without sensors. So it can be affirmed that the use of this system is more effective than taking the standing cast using the STJ neutral position as a reference by probing it, as studies done by McPoil et al. reveal [8,31]. In those studies, when comparing various techniques the standing casts had the worst results of reliability. 

Author such as Michaud [3], describes procedures of taking standing casts but the correction of the foot which is done is verified by probing the head of the talus subjectively. This non-quantifiable verification means that the process may contribute uneven results in the casts. 

Someres et al. [25] state that obtaining favorable results in the technique with plaster casts can be due to the importance of being able to have a good view and perceive the position of the foot, as is reflected in their study.

We consulted a study done by Lapointe [32] that compared the measurement of the calcaneal bisector as a point of reference along with that of the heel to reference the bisector line and value the angulation with the midline of the leg. The results were different, giving greater validity to the measurements done with measuring instruments. Therefore, as this is usually done visually, it is important to exactly outline this bisector if the placing of the heel bone is taken as a reference. This could contribute biases to the impression technique. 

Laughton et al. [24] studied four methods for taking an impression in order to make an orthopedic device: plaster casts standing, half-standing with impressions in phenolic foam and a laser scanning of the foot half-standing and standing. The high variability of the results between the four casting methods can explain the lack of homogeneity and control between the distinct methods. The results suggested that a consequence of the different casting methods was a representation of the foot in different morphological measurements. These differences can affect the comfort, fit and function of the resulting orthosis.

Laser digitalization methods, the same as the technique with plaster casts, result in quite reliable correlation coefficient measurements (ICC).

Guldemond [13] evaluated the results of the orthotic treatment applied to the patient, depending on the type of cast taken. The conclusion was that when taking the cast with phenolic foam the foot’s dimensions to adapt the plantar support are more real and similar to the foot’s clinical dimensions than when taking the cast with plaster cast. Leslie et al. [23] studied the reliability of cast techniques with plaster of Paris and phenolic foam. The method of melting foam had, compared with the plaster method, significantly better intracaster reliability (F = 2.755; *p* = 0.003). Each evaluator’s capacity of measuring the angle of the forefoot with the rearfoot was good. However, the reliability between evaluators was poor. The parameters for the impression of phenolic foam showed significantly less variability than that of plaster: 1.51° compared to 2.46° (*p* = 0.018).

Carroll et al. [9] carried out a study on the reliability of capturing the foot’s parameters using digital scanning and the casting technique in a neutral position. They concluded that the digital exploration was a reliable technique.

Dombroski et al. [28] did a study that revealed the advance of the technique applying digital exploration together with the reproduction of models through 3D printers. Although this is the most exact procedure, the foot’s position to do the digital reading remains unsolved with this system, as the reading can be the same but the foot’s position and the reproducibility can vary for that specific position, as takes place in the aforementioned research. Recent studies support these findings [4,10,14].

## 5. Conclusions

According to the statistical results for the variable “Symmetry index”, the use of pressure sensors verifies a cast centered on the heel and a better plantar support in the horizontal plane (median = 0.0416). This improves the relationship between the plantar contact planes of the forefoot and rearfoot.

Casts taken with sensors maintain a greater symmetry in the curve of the heel valued in the frontal plane with data in the percentile 75. These are below 0.068, indicating a greater control of the foot’s position.

Finally, assistance of pressure sensors when taking casts is reliable, since the result provides a significance value of 0.882 and a confidence level of 90% when we assess the reproducibility of the maneuver.

## Figures and Tables

**Figure 1 sensors-21-07476-f001:**
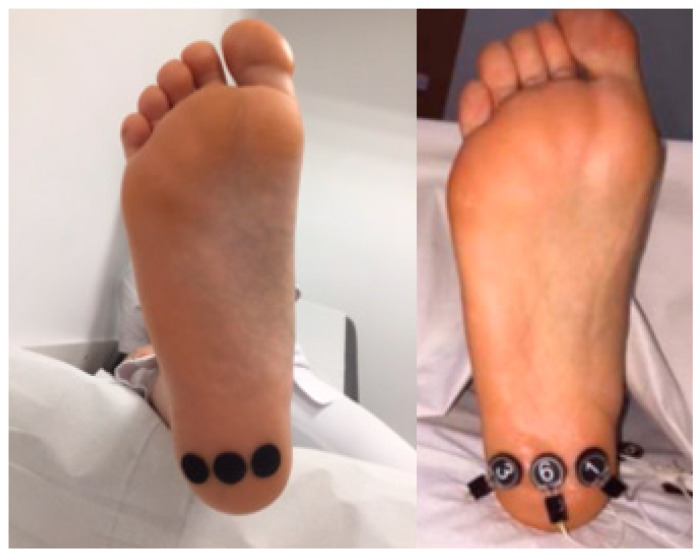
Placing the sensors in the places chosen.

**Figure 2 sensors-21-07476-f002:**
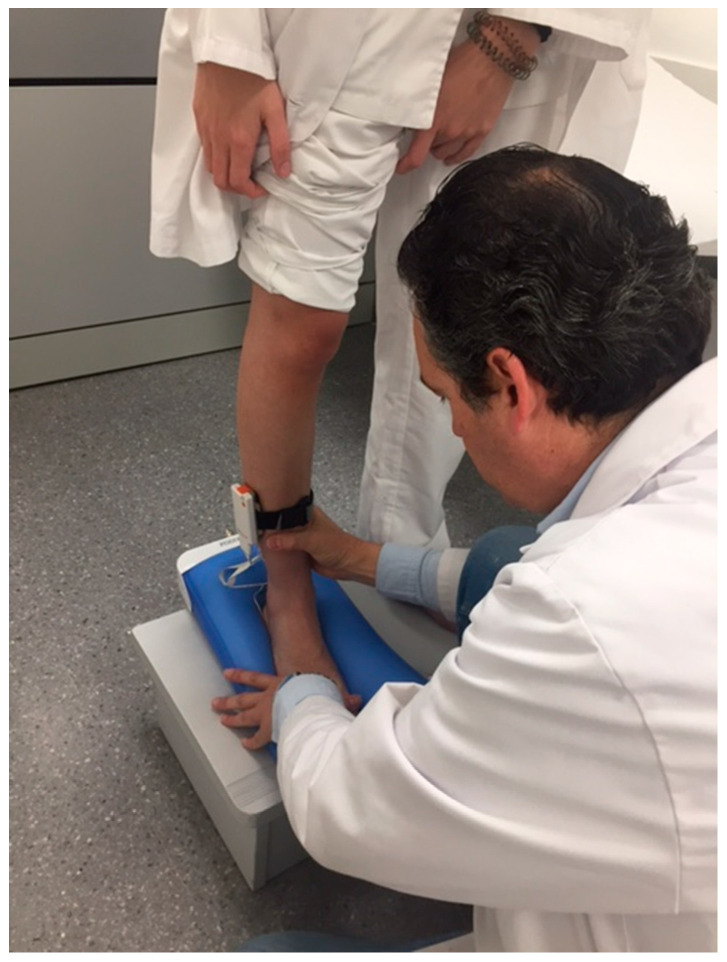
Taking the standing cast with pressure sensors.

**Figure 3 sensors-21-07476-f003:**
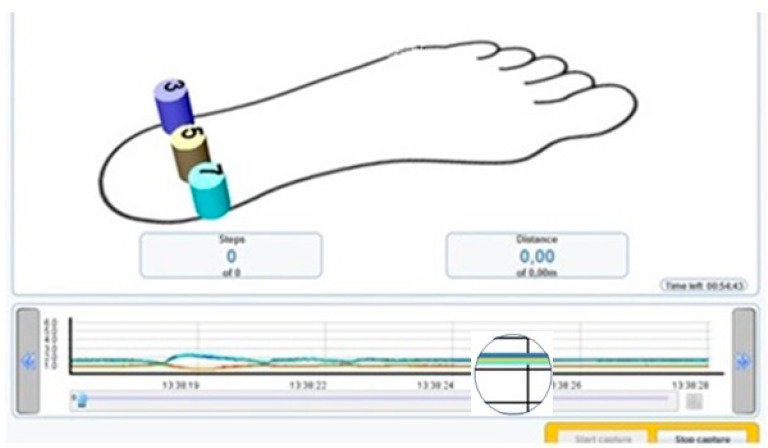
Cast taking moment with sensors and pressure reading, observing on the graph the reading of a single line when the pressures equalize (in that moment, the pressure registered by 3, 5 and 7 sensors are the same).

**Figure 4 sensors-21-07476-f004:**
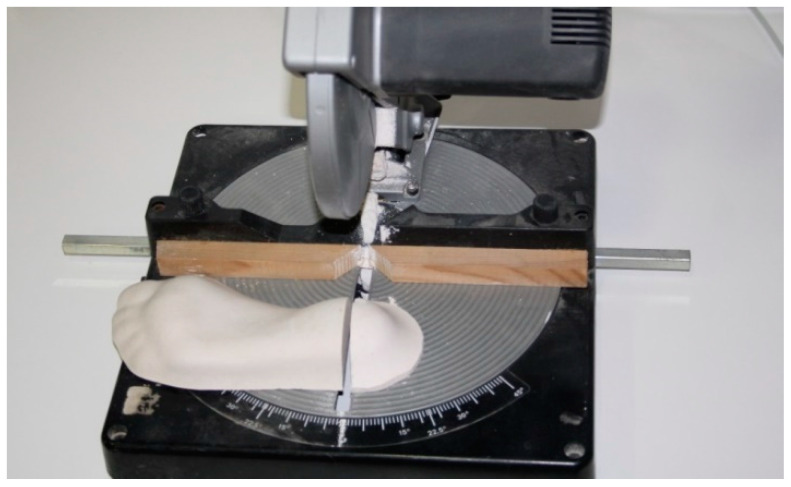
Cut of the cast in the zone of the heel.

**Figure 5 sensors-21-07476-f005:**
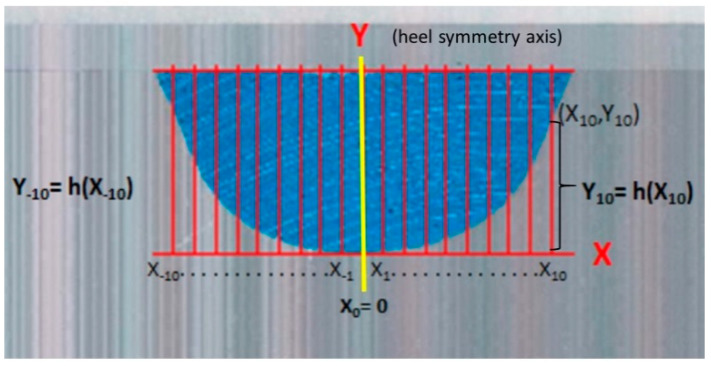
Processing for the measurements of the distinct points of the curve and to determine its symmetry index. 10 coordinates were estimated in each half heel from −10 to +10, beside symmetry “Y” axis (yellow line). For example: (X10, Y10) coordinates were determined measuring the distance of X10 on X axis, and the high of Y10 (h(X10)) on Y axis. For each foot, 20 coordinates were determined, 10 in each half of the heel. The sum of this coordinates is included in the equation.

**Figure 6 sensors-21-07476-f006:**
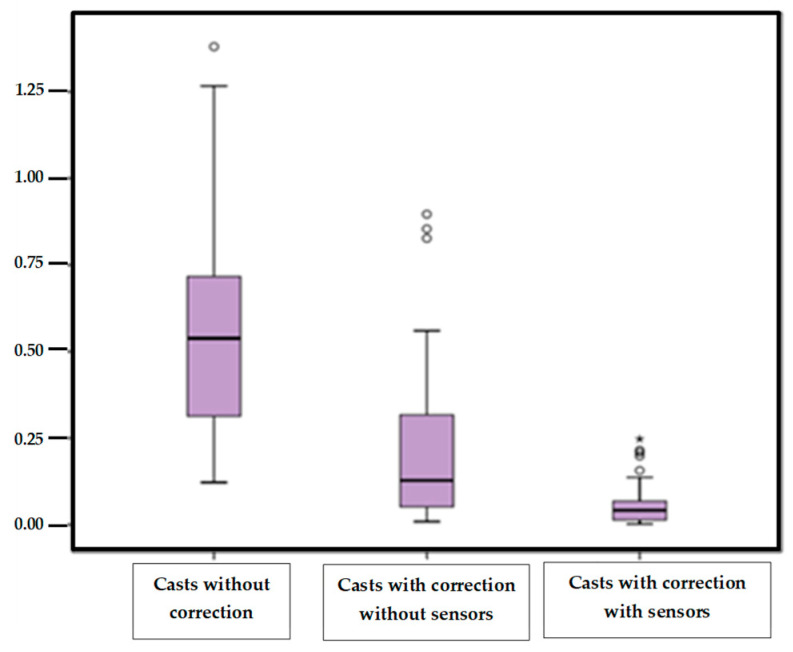
Box plot of the “Symmetry heel” values according to the three variables of the study.

**Table 1 sensors-21-07476-t001:** Descriptive statistics of the “Symmetry Index” in the three study groups.

^1^ Symmetry Index	Casts without Control	Casts without Sensors	Casts with Sensors
*n*	50	50	50
Mean	0.5475	0.2117	0.0562
StandardDeviation	0.2965	0.2258	0.0592
Minimum	0.1222	0.0089	0.0018
Maximum	1.3799	0.8963	0.2472
25 percentile	1.3799	0.0514	0.0135
50 (median)	0.5380	0.1279	0.0416
75 percentile	0.7204	0.3183	0.0685

^1^ Symmetry Index values for the same feet.

**Table 2 sensors-21-07476-t002:** Test of normality.

	^1^ Statistic	^1^ Significance
Casts without correction	0.942	0.016
Casts with correction without sensors	0.803	<0.001
Casts with correction with sensors	0.775	<0.001

^1^ Shapiro–Wilk test. Significance level α = 0.05.

**Table 3 sensors-21-07476-t003:** “Symmetry Index” variable compared by pairs for the three groups.

Sample 1–Sample 2	Friedman Test	Standard Error	Statistical Deviation	*p* Values
Casts with Correction with Sensors/Cast with Correction without Sensors	0.640	0.200	3.200	0.004
Casts with Correction with Sensors/Cast without Correction	1.640	0.200	8.200	<0.001 *
Casts with Correction without Sensors/Cast without Correction	1.000	0.200	5.000	<0.001*

* Significance level α = 0.05.

## Data Availability

The excel data file with all the results is available in the Appendix A section of the journal.

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
