# Peer review of "Use of Plantar Pressure Sensors to Take Weight-Bearing Foot Casts"

_sensors, 2021, doi:10.3390/s21227476_

Round 1

Reviewer 1 Report

lines 6-11: please also mention: city, country. Also please fill in the data: ”Citation: Lastname, F.; Lastname, F.; Lastname, F. Title.” (on page 1, left)

line 13: should be rephrased: e.g.: ”Techniques of taking casts mainly rely not on the objectivity of the procedure, but on the experience and skill of the technician.”

line 20: correct: ”...This enabled the control and quantification...”

line 28: ”...the physical activity of the general population has been progressively increased...” - passive voice: who increased the physical activity of the population? Maybe you should use the active voice („...has progressively...”). Moreover, I think that, on the contrary, with the lockdown and online activities, the physical activity has progressively decreased, not increased.

line 29: correct: ”...therefore, more time dedicated to walking and running becomes absolutely necessary.”

line 46: correct: ”...as compared to the traditional ones described...”

line 50: ”...a subjective technique by the techician” - to be rephrased

line 55: correct: „...the physician”

line 79: please explain: ”... 8 feet were used for internal reliability”

line 146: correct: ”...the measurements shown in Figure 4 were done.”

line 199: correct: ”The Shapiro-Wilk normality test indicated...”

line 209: correct: ”Table 3” (instead of ”Tabla 3”); the last line of the table is not at the same level as the values; significance level: 0.05 (instead of 0,05)

lines 211 and 212: is it the ”reliability of the study” (211), or the ”study of the reliability” (212)?

lines 215-218: ”The Shapiro-Wilk test...”; please explain: ”As a result, non-parametric test was chosen”; ”...in the three measures” = ”in the three measurements” (217-218)?

lines 230-231: ”Comparing different casting techniques of pathology-free structures can affect the information” - how can an information be affected by a comparison?

line 232: ”foot deformation to correct is presented” - please rephrase

line 233: ”differences between the different” - please avoid the repetition

lines 235-236: another repetition: ”it is also influenced by is influenced by”

line 237: correct: ”...model need” (instead of: ”model of need”)

line 244: correct: ”opt for taking” (instead of: ”opt for taken”)

line 262: ”made not standing” - to be rephrased

line 263: correct: ”Somers et al. [22]” (blank space between ”al.” and ”[22]”) - incorrectly cited in the reference as Someres

lines 341-403: all the References should follow the same rules. Here, it is not the case: references 1-4 have the order: authors, year, title etc., whereas references 5-7 have the order: author, title, name of the journal and only afterwards the year; some of them also give the month and others do not; some of them give the DOI as: 10.1002/edm2.132, whereas others give the entire link, as: https://doi.org/10.1186/s12891-019-2898-0; reference 1: ”Prat J.M, ...”, reference 2: ”Levy AE”, reference 7: ”Lee KKW.” etc. etc. Please respect a uniform citation method, as required.

Finally, I think that the experimental part can be further extended.

Author Response

Response to Reviewer 1 Comments

We are grateful to the reviewers for the time and effort they put in reviewing our paper. We are also thankful for the insightful comments, which led us to improve our research. Because we are not English native, we are really grateful for your idiomatic corrections. Thankyou.

Please, find below a response to each comment. For readability purposes, our responses to the comments are highlighted in yellow. We hope that the current version meets the reviewers’ expectations.

Lines 6-11: please also mention: city, country. Also please fill in the data: ”Citation: Lastname, F.; Lastname, F.; Lastname, F. Title.” (on page 1, left)                                                                          They have been completed.

line 13: should be rephrased: e.g.: ”Techniques of taking casts mainly rely not on the objectivity of the procedure, but on the experience and skill of the technician.”                                               It has been modified.

line 20: correct: ”...This enabled the control and quantification...”                                             It has been modified.

line 28: ”...the physical activity of the general population has been progressively increased...” - passive voice: who increased the physical activity of the population? Maybe you should use the active voice („...has progressively...”). Moreover, I think that, on the contrary, with the lockdown and online activities, the physical activity has progressively decreased, not increased. The correction is made.

On the other hand, before the Covid-19 pandemic, in our country, amateur sports and physical activity had increased. We had more sporting events and more participants in running and triathlon competitions. Also, at present, our physicians encouraged the elderly people to walk to keep their quality of life.

line 29: correct: ”...therefore, more time dedicated to walking and running becomes absolutely necessary.”                                                                                                                                                                         It has been modified.

line 46: correct: ”...as compared to the traditional ones described...”                                                        It has been modified.

line 50: ”...a subjective technique by the techician” - to be rephrased.                                                      It has been modified.

line 55: correct: „...the physician”                                                                                                                        It has been modified.

line 79: please explain: ”... 8 feet were used for internal reliability”                                                      The internal reliability was verified using 8 random feet from the total of the sample. To identify the intra-observational variability, three casts of those feet were taken in every three positions.

line 146: correct: ”...the measurements shown in Figure 4 were done.”                                                   It has been corrected.                         

line 199: correct: ”The Shapiro-Wilk normality test indicated...”                                                           The comma has been eliminated.

line 209: correct: ”Table 3” (instead of ”Tabla 3”); the last line of the table is not at the same level as the values; significance level: 0.05 (instead of 0,05)                                                                                 Both have been corrected.

lines 211 and 212: is it the ”reliability of the study” (211), or the ”study of the reliability” (212)? It is “study of the reliability”.

lines 215-218: ”The Shapiro-Wilk test...”; please explain: ”As a result, non-parametric test was chosen”; ”...in the three measures” = ”in the three measurements” (217-218)?                                     The Shapiro-Wilk test verified that the distribution did not meet the normality criteria, so a non-parametric test was chosen.

“In the three measurements”.

lines 230-231: ”Comparing different casting techniques of pathology-free structures can affect the information” - how can an information be affected by a comparison?                                       The phrase has been changed. “The results can be affected when no control is necessary since casting techniques with and without weight bearing are evaluated.”

line 232: ”foot deformation to correct is presented” - please rephrase                                                 The phrase has been modified. “When the foot does not present any deformation to correct…”

line 233: ”differences between the different” - please avoid the repetition                           The phrase has been changed

lines 235-236: another repetition: ”it is also influenced by is influenced by”                                        It has been modified.

line 237: correct: ”...model need” (instead of: ”model of need”)                                                              The “of” word has been eliminated.

line 244: correct: ”opt for taking” (instead of: ”opt for taken”)                                                                    It has been modified.

line 262: ”made not standing” - to be rephrased                                                                  These are more morphologically similar to the casts made with plaster in non-weight-bearing position.

line 263: correct: ”Somers et al. [22]” (blank space between ”al.” and ”[22]”) - incorrectly cited in the reference as Someres.                                                                                                                                          They have been modified.

lines 341-403: all the References should follow the same rules. Here, it is not the case: references 1-4 have the order: authors, year, title etc., whereas references 5-7 have the order: author, title, name of the journal and only afterwards the year; some of them also give the month and others do not; some of them give the DOI as: 10.1002/edm2.132, whereas others give the entire link, as: https://doi.org/10.1186/s12891-019-2898-0; reference 1: ”Prat J.M, ...”, reference 2: ”Levy AE”, reference 7: ”Lee KKW.” etc. etc. Please respect a uniform citation method, as required. References and citation has been changed but because the “track change” is working at manuscript the correct numbers are included in the PDF file.

Finally, I think that the experimental part can be further extended.                                                    More information and also 2 figures have been included.

Reviewer 2 Report

The main idea behind the paper consists in using a pressure sensor for taking standing foot casts. This idea is not a very recent idea, as many researchers have used sensors for correcting foot position. The authors compare three different situations: first in which the subject did not control the position of the foot, second when manipulations corrected the foot’s pronator position and third, using plantar pressure sensors.

The title and the intentions declared in the abstract correspond to the contents of the paper. The authors have some new contributions in this field:

  1. Espinosa-Moyano, I.; Reina-Bueno, M.; Palomo-Toucedo, I.C.; González-López, J.R.; Castillo-López, J.M.; Domínguez-Maldonado, G. Study of the distortion of the indirect angular measurements of the calcaneus due to perspective: In vitro testing. Sensors 2021, 21, doi:10.3390/s21082585.
  2. Reina-Bueno, M.; Vázquez-Bautista, M.D.C.; Palomo-Toucedo, I.C.; Domínguez-Maldonado, G.; Castillo-López, J.M.; Ramos-Ortega, J.; Munuera-Martínez, P.V. Effectiveness of custom-made foot orthoses in patients with systemic lupus erythaematosus: Protocol for a randomised controlled trial. BMJ Open 2021, 11, doi:10.1136/bmjopen-2020-042627.
  3. Rosende-Bautista, C.; Munuera-Martínez, P.V.; Seoane-Pillado, T.; Reina-Bueno, M.; Alonso-Tajes, F.; Pérez-García, S.; Domínguez-Maldonado, G. Relationship of body mass index and footprint morphology to the actual height of the medial longitudinal arch of the foot. Int. J. Environ. Res. Public Health 2021, 18, doi:10.3390/ijerph18189815.
  4. Palomo-Toucedo, I.C.; Leon-Larios, F.; Reina-Bueno, M.; Vázquez-Bautista, M.C.; Munuera-Martínez, P.V.; Domínguez-Maldonado, G. Psychosocial influence of ehlers–danlos syndrome in daily life of patients: A qualitative study. Int. J. Environ. Res. Public Health 2020, 17, 1-14, doi:10.3390/ijerph17176425.
  5. Palomo-Toucedo, I.C.; Vázquez-Bautista, C.; Munuera-Martínez, P.V.; Domínguez-Maldonado, G.; Castillo-López, J.M.; Reina-Bueno, M. Podiatry alterations in Ehlers-Danlos syndrome. Medicina Clinica 2020, 154, 94-97, doi:10.1016/j.medcli.2019.05.006.

The references could be more related to the subject of the paper and are very old (24 titles out of 31 are older than 10 years and 10 of them are older than 20 years). They did not present the novelty of each reference; they only mention these references (example: Page 1, line 40 [2, 3, 8, 10]). So, the Introduction chapter is not really a “state of the art” and I consider that the entire chapter must be rewritten.

In the chapter 2, Materials and Methods, phrase “The Ethical Committee of Experimentation of the University of Seville authorized the research”, page 2, lines 77-78, must be eliminated from the text and put at the end of the paper on the “Institutional Review Board Statement”.

In the same chapter, section Materials, the authors present the addresses of the companies that produce the equipment and materials. They must present the technical characteristics not the postal addresses of the companies.

Page 4, lines 129-130, can the authors present a figure or two that certified the values of pressure with and without correction? I think that it can be more related to the subject than figure 2.

Page 4, lines 129-130, the authors consider that the pressure must be equal on the three sensors?

 Page 5, chapter 3, the first paragraph is ambiguous. How from 58 feet the authors obtained 222 procedures? Can the authors explain this:” to avoid experimental mortality we included 50 ones”?

Page 6, Table 1. Please present all the results in the table. The journal is only online so increasing the number of pages isn’t a problem.

Author Response

Response to Reviewer 2 Comments

We are grateful to the reviewers for the time and effort they put in reviewing our paper. We are also thankful for the insightful comments, which led us to improve our research. Please, find below a response to each comment. For readability purposes, our responses to the comments are highlighted in yellow. Finally, changes in the manuscript are marcked by the “track change”. We hope that the current version meets the reviewers’ expectations.

The main idea behind the paper consists in using a pressure sensor for taking standing foot casts. This idea is not a very recent idea, as many researchers have used sensors for correcting foot position. The authors compare three different situations: first in which the subject did not control the position of the foot, second when manipulations corrected the foot’s pronator position and third, using plantar pressure sensors.

The title and the intentions declared in the abstract correspond to the contents of the paper. The authors have some new contributions in this field:

  1. Espinosa-Moyano, I.; Reina-Bueno, M.; Palomo-Toucedo, I.C.; González-López, J.R.; Castillo-López, J.M.; Domínguez-Maldonado, G. Study of the distortion of the indirect angular measurements of the calcaneus due to perspective: In vitro testing. Sensors 2021, 21, doi:10.3390/s21082585.
  2. Reina-Bueno, M.; Vázquez-Bautista, M.D.C.; Palomo-Toucedo, I.C.; Domínguez-Maldonado, G.; Castillo-López, J.M.; Ramos-Ortega, J.; Munuera-Martínez, P.V. Effectiveness of custom-made foot orthoses in patients with systemic lupus erythaematosus: Protocol for a randomised controlled trial. BMJ Open 2021, 11, doi:10.1136/bmjopen-2020-042627.
  3. Rosende-Bautista, C.; Munuera-Martínez, P.V.; Seoane-Pillado, T.; Reina-Bueno, M.; Alonso-Tajes, F.; Pérez-García, S.; Domínguez-Maldonado, G. Relationship of body mass index and footprint morphology to the actual height of the medial longitudinal arch of the foot. Int. J. Environ. Res. Public Health 2021, 18, doi:10.3390/ijerph18189815.
  4. Palomo-Toucedo, I.C.; Leon-Larios, F.; Reina-Bueno, M.; Vázquez-Bautista, M.C.; Munuera-Martínez, P.V.; Domínguez-Maldonado, G. Psychosocial influence of ehlers–danlos syndrome in daily life of patients: A qualitative study. Int. J. Environ. Res. Public Health 2020, 17, 1-14, doi:10.3390/ijerph17176425.
  5. Palomo-Toucedo, I.C.; Vázquez-Bautista, C.; Munuera-Martínez, P.V.; Domínguez-Maldonado, G.; Castillo-López, J.M.; Reina-Bueno, M. Podiatry alterations in Ehlers-Danlos syndrome. Medicina Clinica 2020, 154, 94-97, doi:10.1016/j.medcli.2019.05.006.

The references could be more related to the subject of the paper and are very old (24 titles out of 31 are older than 10 years and 10 of them are older than 20 years). They did not present the novelty of each reference; they only mention these references (example: Page 1, line 40 [2, 3, 8, 10]). So, the Introduction chapter is not really a “state of the art” and I consider that the entire chapter must be rewritten.                                                                                                     The chapter has been modified. The references and citation has been changed but because the “track change” is working at manuscript the correct numbers are included in the PDF file.

In the chapter 2, Materials and Methods, phrase “The Ethical Committee of Experimentation of the University of Seville authorized the research”, page 2, lines 77-78, must be eliminated from the text and put at the end of the paper on the “Institutional Review Board Statement”.               It has been eliminated.

In the same chapter, section Materials, the authors present the addresses of the companies that produce the equipment and materials. They must present the technical characteristics not the postal addresses of the companies.                                                                                                               They have been eliminated.

Page 4, lines 129-130, can the authors present a figure or two that certified the values of pressure with and without correction? I think that it can be more related to the subject than figure 2. A new figure has been added.

Page 4, lines 129-130, the authors consider that the pressure must be equal on the three sensors? Yes. That is right.

 Page 5, chapter 3, the first paragraph is ambiguous. How from 58 feet the authors obtained 222 procedures? Can the authors explain this:” to avoid experimental mortality we included 50 ones”?                                                         We agree, it is confusing. We have 50 feet of 29 participants who met the inclusion criteria. From the total sample, we take 8 for the internal reliability. That is the final result: 50X3=150 plus 8X3X3=72. We have considered your opinion and reformulated “A total of 150 procedures were performed from 50 feet of 29 volunteers.”…  “To verify the internal reliability of the study, 8 feet were used, so 72 procedures were performed, in addition.”

About “experimental mortality” we refer to broken parts due to fall or deformation when obtaining the positive plaster cast.  This phrase is included “due to falls or deformations when obtaining the positive plaster”.

Page 6, Table 1. Please present all the results in the table. The journal is only online so increasing the number of pages isn’t a problem.                                                                                                                   Since no more variables were included, we believe that all results do not add more information. However, a graphic has been included. An excel file is attached in case you or the editor would consult the data.

Round 2

Reviewer 1 Report

Thank you for responding to all my comments and performing all the corrections that I suggested!

I found in this second version only a few novel aspects that need minor corrections.

Line 35: it should be ”arthritis [5],” instead of ”arthritis, [5]”

Line 36: it should be ”[6]”, instead of ”[6”.

Line 41: blank space between "]" and "." should be removed: ”[8-11] .”

Line 46: full stop missing at the end of the phrase.

Line 58: "[1,]." To be corrected.

Lines 70-71: "walking. [17]This study" to be replaced with "walking
[17]. This study"

In Chapter 2, the titles of the sub-sections should be all in the same manner. You have: "Design and sampling" (in italic); "Recruitment:" (in italic, followed by ":"); "Material" (in italic, with one blank line - line 107 - before); .....; "Taking the Cast in the Group without Correction:" (without italics and with ":") etc.

Line 210: You said that you used 50 feet of 29 volunteers. Maybe it would be good to explain here more clearly why you excluded 8 feet out of the total of 29*2=58 feet and you used only 50.

Author Response

Thank you for your interest to improve the original manuscript. All your suggestions have been incorporated.

A new version has been submitted with the corrections and references without track changes.

Reviewer 2 Report

The authors have improved the quality of the paper. 

Author Response

Thank you for your interest to improve the original manuscript.

A new version has been submitted with some corrections and references without track changes.